# Noradrenaline in Alzheimer’s Disease: A New Potential Therapeutic Target

**DOI:** 10.3390/ijms23116143

**Published:** 2022-05-30

**Authors:** Irene L. Gutiérrez, Cinzia Dello Russo, Fabiana Novellino, Javier R. Caso, Borja García-Bueno, Juan C. Leza, José L. M. Madrigal

**Affiliations:** 1Department of Pharmacology and Toxicology, School of Medicine, Universidad Complutense de Madrid (UCM), Instituto de Investigación Sanitaria Hospital 12 de Octubre (Imas12), Instituto de Investigación Neuroquímica (IUINQ-UCM), Centro de Investigación Biomédica en Red de Salud Mental (CIBERSAM), Avda. Complutense s/n, 28040 Madrid, Spain; irenlo02@ucm.es (I.L.G.); fabnovel@ucm.es (F.N.); jrcaso@med.ucm.es (J.R.C.); bgbueno@med.ucm.es (B.G.-B.); jcleza@med.ucm.es (J.C.L.); 2Department of Healthcare Surveillance and Bioethics, Section of Pharmacology, Università Cattolica del Sacro Cuore, Fondazione Policlinico Universitario A. Gemelli IRCCS, 00168 Rome, Italy; cinzia.dellorusso@unicatt.it; 3Department of Pharmacology and Therapeutics, Institute of Systems, Molecular and Integrative Biology (ISMIB), University of Liverpool, Liverpool L69 3GL, UK; 4Institute of Molecular Bioimaging and Physiology (IBFM), National Research Council, 88100 Catanzaro, Italy

**Keywords:** noradrenaline, norepinephrine, locus coeruleus, Alzheimer’s disease, neuroinflammation, inflammation, cytokines, glia

## Abstract

A growing body of evidence demonstrates the important role of the noradrenergic system in the pathogenesis of many neurodegenerative processes, especially Alzheimer’s disease, due to its ability to control glial activation and chemokine production resulting in anti-inflammatory and neuroprotective effects. Noradrenaline involvement in this disease was first proposed after finding deficits of noradrenergic neurons in the locus coeruleus from Alzheimer’s disease patients. Based on this, it has been hypothesized that the early loss of noradrenergic projections and the subsequent reduction of noradrenaline brain levels contribute to cognitive dysfunctions and the progression of neurodegeneration. Several studies have focused on analyzing the role of noradrenaline in the development and progression of Alzheimer’s disease. In this review we summarize some of the most relevant data describing the alterations of the noradrenergic system normally occurring in Alzheimer’s disease as well as experimental studies in which noradrenaline concentration was modified in order to further analyze how these alterations affect the behavior and viability of different nervous cells. The combination of the different studies here presented suggests that the maintenance of adequate noradrenaline levels in the central nervous system constitutes a key factor of the endogenous defense systems that help prevent or delay the development of Alzheimer’s disease. For this reason, the use of noradrenaline modulating drugs is proposed as an interesting alternative therapeutic option for Alzheimer’s disease.

## 1. Introduction

Noradrenaline (NA), dopamine and adrenaline constitute a group of endogenous substances known as catecholamines. They receive this name because they all are amines attached to a catechol group. NA and adrenaline are also called norepinephrine and epinephrine, respectively, these names being derived from the Greek *epi* and *nephros* “on/over kidneys” or from Latin *ad* and *rene* “at kidneys”. Therefore, the name adrenaline refers to the adrenal glands because most of the adrenaline present in our bodies is produced and stored in these glands. NA, while it is also produced at relatively small concentrations in the adrenal medulla and can function as a hormone, is mainly produced by sympathetic neurons in the central nervous system (CNS) where it acts as a neurotransmitter [1].

The specialized neurons that produce NA are clustered in small nuclei located in the pons and lateral reticular formation of the medulla. Among these nuclei, the locus coeruleus (LC) is responsible for the production of most of the NA found in the brain [2]. The projections stemmed from the LC noradrenergic cells are highly ramified, enabling distribution of NA through most brain areas. In addition, NA can be released extrasynaptically by axonal varicosities near glial cells and micro vessels controlling their activity.

Once released from neuronal terminals NA can activate specific adrenergic receptors (αAR and βAR) and generate its best-known effects, those associated with the “fight or flight” response. However, additional effects, of NA in the regulation of other processes that take place in the CNS have also been described during the last years. NA is a major modulator of behavior controlling different cognitive functions such as attention, motivation, stress, sleep, mood, and memory among others [3,4,5,6]. NA also modulates neuronal metabolism, synaptic plasticity, glial cells activity, and blood-brain barrier permeability [7]. In addition, NA plays a key neuroprotective role in neuroinflammatory processes [8]. Knowledge of this action of NA arose from the initial postmortem analyses of brains from Alzheimer’s disease (AD) [9,10] and Parkinson´s [11] disease patients in which a reduction in the number of neurons from the LC was observed. These seminal observations have been confirmed in recent years, showing a relationship between the degree of disease progression and the numbers of LC neurons lost in the LC. In line with the postmortem observations that connect the degeneration of noradrenergic neurons and the progression of AD according to clinical assessments, the association between the loss of LC neurons and the worsening of antemortem global cognitive function, memory loss and visuospatial ability has been demonstrated [12].

However, despite the evidence of LC degeneration in AD, its contribution to pathology was underappreciated for years. Recently, several reports have studied the role of the noradrenergic system in AD and highlighted its potential as a therapeutic target [13,14,15,16]. Nevertheless, the role of the noradrenergic system in AD remains unclear, and controversial studies about NA pointed out both to neuroprotective and neurotoxic actions in AD. Thus, an exhaustive analysis of the literature could provide new insights to disentangle the role of the noradrenergic system dysfunction in AD.

The purpose of this review is to recapitulate the available evidence about the NA molecular mechanisms of action and the noradrenergic system in AD pathogenesis. Furthermore, possible detrimental effects of NA and future research directions will be also explored. A summary of the main findings is presented in graphical abstract.

## 2. Dysfunction of the Noradrenergic System in Alzheimer’s Disease

AD is an incurable neurodegenerative disease whose prevalence is expected to increase exponentially in the coming years [17]. Clinically, it is characterized by a progressive decline in memory, language, and other cognitive functions. These cognitive deficits are consequences of neuronal loss probably related to the accumulation of intracellular inclusions of aberrant forms of phosphorylated tau and extracellular deposits of amyloid-β (Aβ), known as neurofibrillary tangles (NFTs) and amyloid or senile plaques, respectively.

AD is diagnosed and treated when clinical symptoms become apparent, and the neuronal dysfunction is established and irreversible. Therefore, most of the therapeutic strategies developed in the last years are ineffective. So, understanding the “preclinical” phase of AD could provide an optimal time window for early initiation of therapeutic interventions, thus potentially increasing their efficacy.

### 2.1. Locus Coeruleus Degeneration in Alzheimer’s Disease

In the early 1980s, it was described for the first time that LC degenerates early in AD before any other pathological feature appears, and later studies supported this discovery [9,10,18]. The LC neuronal loss correlated better with the duration of illness than that found in the nucleus basalis (NB) as traditionally described [19]. Furthermore, before noradrenergic neuronal death occurs, there is a decrease in the large pigmented neurons of the LC due to cell shrinkage and a loss of phenotype [18].

Postmortem analysis also showed that LC is one of the first brain regions where Aβ is deposited [20], and Braak and colleagues characterized the LC as the first site where NFTs start to accumulate [21], even before the age of thirty [22,23].

The use of neuroimaging techniques, such as the Magnetic Resonance Imaging (MRI) and Positron Emission Tomography (PET), allows for the in vivo study of the LC in the human brain. These techniques offer complementary information. Therefore, studies with different approaches have been conducted in recent years (Table 1).

MRI-based imaging can provide both structural and functional information on LC with high spatial resolution and good reliability in a safe and non-invasive manner. By exploiting the paramagnetic properties of neuromelanin (NM)-rich neurons in the LC, several structural MRI studies allowed the quantification of both LC size and NM content, which can be considered a measure of LC integrity.

The atrophy of LC was detected in mild cognitive impairment (MCI) and AD patients [24]. Remarkably, a reduction of LC volume in initially asymptomatic healthy controls was associated with the development of AD within two years [24]. LC atrophy correlated with the amyloid level in the cerebrospinal fluid in AD [25]. LC size also was an indicator of cognitive reserve in control subjects, MCI and AD patients, with a progressive reduction of LC volume among these three groups in parallel to their cognitive performance [26]. Moreover, the NM signal in LC was reduced in AD subjects and the extent of such reduction correlated to the severity of neuropsychiatric symptoms [27]. Therefore, the existence of noradrenergic alterations in the earlier stages of AD seems to be a well-established factor that favors the progression of this disease. For this reason, considering the relevance of the NA pathway, the LC degeneration can be taken as an alternative and earlier biomarker of AD. This would allow the development of alternative therapeutic strategies based on the use of NA modulating drugs in a time point at which the neurological damage may not be too disabling and a benefit for patients can be achieved far more easily than at a later time when the restoration of NA levels may not have such an impact on neuronal viability [14].

Functional MRI also provided evidence of the key role of LC in brain organization. Indeed, LC was involved in orienting and responding to external stimuli, facilitating the engagement of the executive control systems [28]. Moreover, individuals with parental history of AD displayed a reduction of functional connectivity of LC with several regions of the brain. This finding was closely related to the deterioration in memory capacity, thus demonstrating that LC is a crucial region for AD vulnerability in this at-risk population [29].

Based on a specific radiotracer, PET imaging gives information about brain metabolism, occurrence of neuroinflammation, or deposition of aggregates typical of neurodegenerative disease, including amyloid and tau neurofibrillary tangles in AD. Using a tracer able to measure catecholamine synthesis capacity showed that a decrease in LC catecholamine production was associated with a reduction in memory performance in healthy older adults [30]. PET has also been used in association with MRI, demonstrating the relationship between the LC deterioration, amyloid-beta deposition and tau pathology in AD patients [31], in subjects carrying the mutation for autosomal-dominant AD [32] and in asymptomatic subjects developing cognitive decline over time [33] (Table 1).

Taken together, post-mortem and in vivo evidence demonstrates that LC is strongly involved in early AD-related pathology, also in preclinical and prodromal stages, and that its progressive deterioration correlates with the severity of the disease. In vivo evidence further suggests the LC relevance in modulating the flow of cognitive functions in the human brain and that LC dysfunction has important implications for AD development. Therefore, the evaluation of LC size by MRI could represent a potential in vivo biomarker for AD diagnosis and monitoring of disease progression and a screening biomarker for selecting high-yield candidates to undergo clinical trials.

### 2.2. NA Levels in Alzheimer’s Disease

The loss of noradrenergic neurons in the LC leads to a reduction of the brain levels of NA with correlates with cognitive impairment [34] (Table 1). In addition, the levels of NA in plasma correlate with AD progression and could be a potential biomarker to follow the disease evolution [35]. Thus, elevating NA signaling could provide a new pharmacological target to delay AD progression.

However, to compensate for the LC neurons’ death and lower levels of NA in the brain, the noradrenergic system undergoes different changes [36]. In AD patients with severe LC degeneration, the surviving LC neurons increase the expression of the enzymes involved in the NA synthesis such as thyroxine hydrolase (TH), and reduce the NA uptake at synaptic terminals resulting in increased levels of NA in the cerebrospinal fluid (CSF) which correlated with poor cognitive function [37]. In addition, the surviving neurons from the LC increase the number of axonal and dendritic connections to retain the number of contacts [38,39]. These compensatory changes in the noradrenergic system allow maintainance of adequate NA signaling despite the LC loss.

### 2.3. Adrenergic Receptors in Alzheimer’s Disease

The reduction of brain NA levels also leads to compensatory changes in adrenergic receptors. β_1_AR and β_2_AR density is elevated in the hippocampus and the frontal cortex [34]. Regarding α-adrenergic receptors, one study showed a decrease of α_1A_AR and α_1D_AR in the hippocampus from dementia patients, and α_2A_AR in the prefrontal cortex [39] (Table 1).

These changes in the adrenergic receptors could compensate the changes in NA levels in the early stages of AD, but also contribute to Aβ and tau burden in later stages, favoring AD progression. Studies using β2 agonists show higher levels of amyloid plaque due to γ–secretase activation [40]. This way, the upregulation of β_2_AR may contribute to Aβ deposition in AD pathogenesis. Thus, the dysfunction of adrenergic receptors, rather than a mere loss of LC neurons and brain NA levels, could also be a key contributor to AD pathogenesis.

**Table 1 ijms-23-06143-t001:** Dysfunction of the noradrenergic system in Alzheimer’s disease.

Authors and Year of Publication [Ref.]	Cohort	Study Type/Sample	Main Results of the Study
** *LC degeneration* **			
Cassidy et al., 2022 [27]	C (*n* = 118) MCI (*n* = 44) AD (*n* = 28)	MRI	Loss of neuromelanin signal in LC that predicted neuropsychological symptoms, independently of Aβ and tau burden
Ciampa et al., 2022 [30]	C (*n* = 49)	PET	LC catecholamine synthesis capacity associated with better memory performance
Dahl et al., 2022 [32]	ADAD (*n* = 9)	MRI and PET	LC size associated with tau burden and memory decline
Prokopiou et al., 2022 [33]	C (*n* = 128)	MRI and PET	LC activity associated with Aβ related cognitive decline
Plini et al., 2021 [26]	C (*n* = 395) MCI (*n* = 156) AD (*n* = 135)	MRI	LC size predicted attention and brain maintenance in AD, MCI and controls
Bolton et al., 2021 [41]	sEOAD (*n* = 115) LOAD (*n* = 672)	Postmortem analysis	LC atrophy correlated with poor attentional functioning
Jacobs et al., 2021 [31]	AD (*n* = 221)	MRI and PET	Tau accumulation in LC associated to disease progression
Dutt et al., 2020 [24]	C (*n* = 814) MCI (*n* = 542) AD (*n* = 160)	MRI	LC atrophy in AD, MCI and controls that developed dementia
Del Cerro et al., 2020 [29]	O-LOAD (*n* = 31) C (*n* = 28)	MRI	Decreased LC functional connectivity was correlated with poor memory performances in O-LOAD
Betts et al., 2019 [25]	C (*n* = 25) SCD (*n* = 21) MCI (*n* = 16) AD (*n* = 11)	MRI	LC volume decreased in AD patients and correlated with CSF amyloid levels
Eser et al., 2018 [42]	C (*n* = 3) AD (*n* = 14) CBD (*n* = 14) PSP (*n* = 10)	Postmortem analysis	LC neurons decreased in AD
Theofilas et al., 2018 [43]	AD (*n* = 24)	Postmortem analysis	NFTs accumulation in LC associated with neuronal loss
Theofilas et al., 2017 [44]	AD (*n* = 68)	Postmortem analysis	LC neuronal loss associated with AD progression
Kelly et al., 2017 [12]	C (*n* = 11) MCI (*n* = 10) AD (*n* = 8)	Postmortem analysis	LC neuronal loss correlates better with illness duration
Zhang et al., 2016 [28]	C (*n* = 250)	MRI	LC activation in orienting and sensorimotor responses to external stimuli
Gulyás et al., 2010 [45]	C (*n* = 10) AD (*n* = 10)	Postmortem analysis	NET density decreases in AD
Lyness et al., 2003 [46]	67 studies	Meta-analysis	LC degenerated in AD
German et al., 1992 [47]	C (*n* = 7) PD (*n* = 6) AD (*n* = 9) DS (*n* = 3)	Postmortem analysis	LC neuronal loss correlated with AD duration
Bondareff et al., 1981 [9]	C (*n* = 10) AD (*n* = 20)	Postmortem analysis	LC neuronal loss in AD
** *NA levels* **			
Pillet et al., 2020 [35]	C (*n* = 17) OD (*n* = 22) AD (*n* = 32)	Plasma	↑ NA
Matthews et al., 2002 [34]	C (*n* = 33) AD (*n* = 46)	Temporal lobe	↓ NA
Elrod et al., 1997 [37]	C (*n* = 96) AD (*n* = 74)	CSF	↑ NA
** *Adrenergic receptors* **			
Szot et al., 2007 [39]	C (*n* = 17) AD (*n* = 15)	HP	↓ α_1A_AR and α_1D_AR
Matthews et al., 2002 [34]	C (*n* = 33) AD (*n* = 46)	HP and PFC	↑ β_1_AR and β_2_AR
Szot et al., 2007 [39]	C (*n* = 17) AD (*n* = 15)	PFC	↓ α_2A_AR
Matthews et al., 2002 [34]	C (*n* = 33) AD (*n* = 46)	HP	↔ α_2A_AR

LC: locus coeruleus, NA: noradrenaline, C: control, AD: Alzheimer’s disease, MCI: mild cognitive impairment, OD: Other dementia, ADAD: autosomal-dominant Alzheimer’s disease, sEOAD: early onset Alzheimer’s disease, LOAD: late onset Alzheimer’s disease, SCD: subjective cognitive decline, CDB: corticobasal degeneration, PSP: progressive supranuclear palsy, PD: Parkinson disease, DS: Downs syndrome, MRI: magnetic resonance image, PET: positron emission tomography, CSF: cerebrospinal fluid, HP: hippocampus, PFC: prefrontal cortex.

## 3. Locus Coeruleus Vulnerability to Degeneration

The early loss of LC neurons in neurodegenerative diseases suggests that noradrenergic neurons are more vulnerable than other neuronal types to neuropathologic changes [13]. Research has described how noradrenergic neurons are especially vulnerable to degeneration due to their higher exposure to environmental toxins, oxidative stress, and noradrenergic toxic metabolites [13].

### 3.1. Environmental Toxins

Noradrenaline released by the LC terminals controls the blood flow in the brain and maintains the blood-brain barrier function and integrity. Noradrenergic neurons innervate most of the microvasculature of the brain. Hence, blood-brain barrier dysfunction can increase the exposure of noradrenergic terminals to blood toxins. In addition, the LC is near to the wall of the fourth ventricle and more exposed to toxins in cerebrospinal fluid. The exposure of noradrenergic neurons to environmental toxins can damage them and contribute to their dysfunction and degeneration (Fiure 2). In addition, the accumulation of heavy metals such as silver or mercury can trigger tau accumulation in LC neurons and their subsequent neurodegeneration [48]. Furthermore, noradrenergic axons are poorly or incompletely myelinated [49]. The lack of myelin further contributes to exposing the axon to toxins and other pathogenic molecules.

### 3.2. Oxidative Stress

Oxidative stress is another factor that contributes to the vulnerability of LC neurons. These cells exhibit autonomous activity accompanied by oscillations in Ca^+2^ that elevate mitochondrial oxidative stress contributing to their degeneration [50]. Morphologically, noradrenergic neurons in the LC have disproportionally long and thin axons [49]. Thus, high energy cost is required to preserve the efficiency of synaptic conductance along the axons leading to higher susceptibility to oxidative stress. The higher oxidative status makes the LC neurons more vulnerable to chronic neuroinflammation damage through the activation of the superoxide-generating enzyme NADPH oxidase (NOX2) [51].

### 3.3. Noradrenergic Toxic Metabolites

Recent research suggests that noradrenergic metabolites can also contribute to LC vulnerability and neurodegeneration. In the CNS, NA is mainly degraded by monoamine oxidase A (MAO-A) and catechol-O-methyl transferase. Other enzymes that also play an important role in NA metabolism are aldehyde reductase and aldehyde dehydrogenase (AD) (Figure 1).

MAO-A oxidizes NA into the toxic metabolite 3,4-dihydroxyphenyl glycolaldehyde (DOPEGAL). The latter plays an important role in the activation of tau aggregation in the LC [52]. It was recently demonstrated that the ε4 allele of the ApoE protein (ApoE4) can induce NA oxidation into DOPEGAL [53]. It is well described that ApoE4 increases the risk for AD. However, so far no relationship has been established between ApoE4 and LC vulnerability. In this study, it was demonstrated that ApoE4 binds to the vesicular monoamine transporter 2 (VMAT2) inhibiting noradrenaline uptake. The leakage of NA from synaptic vesicles leads to its oxidation into the metabolite DOPEGAL. This toxic metabolite activates the asparagine endopeptidase (AEP) that cleaves tau and triggers LC neurodegeneration. In addition, DOPEGAL can also react directly with the Lys353 residue of tau stimulating its aggregation and propagation [54].

Aldehyde reductase and catechol-O-methyl transferase can metabolize DOPEGAL into the metabolite 3-methoxy-4- hydroxyphenylethyleneglycol (MHPG). Higher levels of MHPG were associated with lower cortical thickness [55], disease progression and greater cerebrospinal concentrations of p-tau and Aβ [56].

Another molecule that could contribute to LC degeneration is NM. During brain aging NM is accumulated inside neurons acting as a quencher for toxic molecules but also contributing to the pathogenesis of neurodegenerative diseases [57,58]. After neuronal degeneration, NM is released with other toxic molecules activating the immune system and exacerbating the oxidative stress [59]. The LC and the substantia nigra (SN) constitute the brain areas with the highest concentration of NM [60]. In 2015, the mechanism through which NA and its metabolites can promote the synthesis of NM was described [58].

Thus, multiple factors contribute to the vulnerability of LC neurons, although further research is still needed. Understanding the mechanism leading to LC degeneration will provide the necessary evidence to find new therapeutic targets to prevent or delay LC neuronal loss. These treatments can potentially be more effective through targeting the early stages of the disease before irreversible neuronal damage occurs and clinical symptoms become apparent.

## 4. Reduction of LC and NA Activity: Evidence from Preclinical Models

The noradrenergic neuronal degeneration found in AD patients was also present in the animal models used to study AD. It was demonstrated that degeneration of LC in APP/PS1 mice correlates with neuroinflammation and microglia activation [61]. In addition, early hyperphosphorylated tau deposition in LC was found in a transgenic rat model of AD [62].

Other studies focused on the potential connection between the damage to the LC (and subsequent reduction of brain NA levels) and the progression of AD. This has generated pre-clinical data supporting the hypothesis that reduction in brain NA levels by LC loss contributes to AD progression. A summary of the most relevant literature and results is provided in Table 2.

### 4.1. DSP4

One of the initial strategies used to reproduce the loss of LC neurons is the injection of DSP4 (N-(2-chloroethyl)-N-ethyl-2-bromobenzylamine), a selective neurotoxin for birds and rodents that is relatively simple to utilize. After its intraperitoneal injection, DSP4 crosses the blood-brain barrier and is converted into a derivate that can be transported inside the noradrenergic axons by the NA transporter. Once inside these neurons, the DAP4 metabolite causes the destruction of the neuronal terminals [63]. However, not all brain areas are affected equally; the hypothalamus, mesencephalon and pons-medulla show a greater reduction of NA levels than other areas like the cortex, hippocampus, cerebellum, or spinal cord. In addition, while most neurons from the LC are affected by DSP4, the fibers originating in A1 and A2 cell groups are not modified [64]. As a result, the production and distribution of central NA are rapidly reduced. This way, the administration of DSP4 can reproduce some of the alterations resulting from the LC degeneration known to occur in AD.

When the effect of DSP4 was analyzed in brain cortices from rats that received amyloid beta (Aβ) through direct bilateral intracortical injections, an earlier, more robust and longer inflammatory response was observed. This was characterized by the production of inflammatory cytokines interleukin 1β (IL-1β) or IL6 as well as other mediators such as GFAP, NOS2 or COX2 [65]. In particular, in DSP4 treated animals the overexpression of NOS2 was detected primarily at neuronal level, an alteration also observed in the AD brain [66].

Subsequent studies were performed using genetic alterations to cause Aβ pathology, such as that found in amyloid precursor protein 23 (APP23) mice. In this case, the administration of DSP4 to APP23 mice proved to have detectable effects 6 months later. These included not only increased neuroinflammation, but also amyloid plaques accumulation, neuronal loss and altered neuronal metabolism. Behavioral alterations commonly considered to reproduce AD symptoms such as memory deficits were also found to be increased as a consequence of DSP4 treatment [67]. Similarly, it was found later that DSP4 administration results in the inhibition of the Aβ degrading enzyme neprilysin in another murine model of AD also based on the overexpression of human APP [68]. In fact, DSP4 treated AAP-transgenic mice show larger accumulations of Aβ which seems to be the consequence of a reduced microglial recruitment to the Aβ plaque sites and impaired Aβ phagocytosis [69]. Studies performed in mice carrying APP and presenilin mutant transgenes (APP/PS1 mice) also showed increases in inflammatory markers as a result of DSP-4 injections [70,71,72].

The effects of DSP4 have also been evaluated in the P301S mouse model based on the increased expression of mutant human tau protein. In this case, DSP4 treated mice displayed increased memory deficits, neuroinflammation, neurodegeneration and mortality [73], thus confirming that the loss of LC neurons potentiates the alterations caused by the two main agents known to contribute to the progression of AD, namely Aβ and NFTs.

### 4.2. β-Hydroxylase Knockout Mice

The use of DSP4 has certain disadvantages as the drug suppresses not only NA, but also other neuromodulators produced in the LC such as galanin or brain-derived neurotrophic factor (BDNF), which have also been implicated in AD [74,75]. Besides, the injury to neurons caused by DSP4 could itself promote inflammatory responses. Indeed, it has recently been shown that intracerebroventricular administration of DSP4 increases microglial and astrocyte activation, reducing the learning abilities of the animals as well as their memory. However, DSP4 treatment resulted in protective effects against hippocampal dependent learning and memory impairment due to surgery in a model of post operative cognitive dysfunction [76]. For this reason, the development of alternative ways to simulate the depletion of LC neurons could yield more reliable results. Among these, the genetic removal of dopamine β-hydroxylase, the enzyme responsible for the conversion of dopamine to NA, constitutes an interesting strategy as it results in the suppression of NA production but does not affect other co-transmitters or neuronal integrity. In fact, the memory deficits observed in mice overexpressing mutant APP and presenilin-1 are increased if these mice are crossed with dopamine β-hydroxylase knockout mice [77].

### 4.3. Ear2 Knockout Mice

Strategies aiming to prevent the development of LC neurons, lead to similar results. In particular, mice lacking the nuclear receptor Ear2, that is required for the development of LC neurons in mice, display neuronal deficits at this level. Consistently, crossing these mice with those overexpressing mutant APP and presenilin-1, also demonstrated that this alternative depletion of LC neurons potentiates memory deficits as well as long-term potentiation impairment [78].

### 4.4. Beta-Blockers

Alternatively, in order to reproduce the changes that occur in the CNS after the loss of LC neurons, without altering the viability of these cells and their production of NA, it is possible to simply reduce NA activity. This can be achieved pharmacologically by blocking some of the specific adrenergic receptors. There are several antagonists available for the different types of adrenergic receptors, and the most relevant ones due to their widespread use are those commonly known as beta blockers. These drugs are specific antagonists for the β type of adrenergic receptors and are indicated to reduce blood pressure and to treat cardiovascular diseases. Notably, when transgenic APP mice were treated with the beta blocker metoprolol, an increase in the expression of neuroinflammation markers could be detected in combination with impaired cognitive behavior [79]. However, the clinical use of β-blockers has been associated with protective effects in humans with respect to development of AD, as discussed below.

**Table 2 ijms-23-06143-t002:** Dysfunction of noradrenergic system in mouse model.

Authors and Year of Publication [Ref.]	Animal Model	Main Results of the Study
** *DSP-4* **		
Chalermpalanupap et al., 2018 [73]	P301S Tg mice	Increased memory deficits, inflammation, neurodegeneration and mortality
Jardanhazi-Kurutz et al., 2011 [70]	APP/PS1 Tg mice	Increased inflammation and amyloid plaques
Jardanhazi-Kurutz et al., 2010 [71]	APP/PS1 Tg mice	Increased amyloid plaques
Heneka et al., 2010 [69]	APP V717F Tg mice	Increased amyloid plaques and glial activation
Pugh et al., 2007 [72]	APP/PS1 Tg mice	Increased inflammation
Kalinin et al., 2007 [68]	APP V717F Tg mice	Increased amyloid plaques and glial activation
Heneka et al., 2006 [67]	APP23 Tg mice	Increased inflammation, amyloid plaques and neuronal loss
Heneka et al., 2002 [65]	Rats injected with Aβ	Increased inflammation
** *β-Hydrolase knockout mice* **		
Hammerschmidt et al., 2013 [77]	APP/PS1 Tg mice	Increased memory deficits
** *Erk2 knockout mice* **		
Kummer et al., 2014 [78]	APP/PS1 Tg mice	Increased memory deficits
** *β-blockers* **		
Evans et al., 2020 [79]	APP Tg mice	Increased inflammation and deficits in cognitive behaviour

## 5. Direct Effects of Noradrenaline

As mentioned above, according to postmortem observations and the results obtained in the animal models used, it is widely accepted that the reduction of brain NA levels derived from the loss of LC neurons could contribute to the development and/or progression of AD. However, a reasonable way to confirm that NA depletion has these negative effects is to test if NA supplementation elicits the opposite ones. In this regard, in vitro and in vivo studies have been performed, confirming the involvement of LC loss in AD and providing also new insights into the neuroprotective actions of NA.

### 5.1. Anti-Inflammatory Actions

The anti-inflammatory actions of NA are considered part of the mechanisms that make the brain an immune-privileged organ. This has been proposed based on the inhibition of the interferon gamma (IFNγ) dependent upregulation of major histocompatibility complex class II antigens (MHC-II) in cultured rat astrocytes induced by NA. The inhibitory effect of NA was mediated by the activation of β2-adrenergic receptors and the following production of the second messenger cAMP [80], together with the involvement of protein kinase C (PKC) downstream pathways [81]. Consistently, this was also demonstrated to be the mechanism through which NA prevents the production of the inducible enzyme NOS2 in similarly cultured astrocytes [82]. In addition, NE was observed to reduce the expression of a NOS2 reporter gene, suggesting that the inhibition of NOS2 is due to some modification of the transcription factor activity at the NOS2 gene promoter [83]. Furthermore, a specific region located between bases −187 to −160 of the promoter was found to be critical for the mediation of NA suppressive effects on NOS2 expression [84].

The actions of NA observed at the transcription level led researchers to analyze the potential regulation of certain transcription factors involved in the neuroinflammatory processes. Among these, NFκB is one of the most relevant factors due to its key regulatory role for numerous genes involved in the progression of inflammation. NFκB enters the cell nucleus and binds to specific DNA sequences to regulate gene transcription. This is a relatively fast process because it does not require the new synthesis of NFκB, but only its activation. Indeed, NFκB is normally present in the cytoplasm in an inactive state. The inactivation depends on its link to IκBα, a protein that masks the nuclear localization signals of NFκB keeping it inactive. Once IκBα is degraded, NFκB can enter the cell nucleus and modulate gene expression. This pathway was demonstrated to be controlled, at least in part by NA. In primary cortical astrocytes, NA was found to increase the expression of IκBα through the activation of β adrenergic receptors in cortical astrocytes [85]. Consistently, DSP4-induced LC degradation resulted in a reduction of brain frontal cortex levels of IκBα [65,85].

In addition to astrocytes, microglial cells play a key role in the regulation of the neuroinflammatory response. Microglia are unique myeloid cells scattered throughout the CNS parenchymal tissue, representing the first line of immune defense [86]. Through continuous monitoring of the CNS microenvironment, microglia ensure the brain homeostasis, although they contribute to the regulation of other relevant functions in the CNS from early development throughout the adult life [87]. These include, for example, the development and maintenance of the synaptic architecture and function, synaptic pruning, neurogenesis, neuronal activity and plasticity as well as the secretion of neurotrophic factors [88]. Therefore, microglia dysfunctions appear to be a critical factor for the development of neurological disorders, including AD.

Microglia represent the main source of inflammatory mediators that are known to contribute to amplify the damage signals [89]. Therefore, NA inhibition of microglial activation is considered to have a larger impact on the reduction of the neuronal damage resulting from an uncontrolled glial response. For this reason, the effects of NA on primary microglial cultures were analyzed. It has been shown that rat microglial cells express different subtypes of adrenergic receptors, namely α1A, α2A, β1 and β2 receptors [90]. Expression of functional β1 and β2 adrenergic receptors on rat microglial cells was also confirmed in a different study [91]. Treatment of these cells with NA, as well as with the β2 receptor agonist terbutaline markedly increased the intracellular levels of cAMP, whereas the β1 agonist dobutamine and the α1 agonist phenylephrine were less effective, thus suggesting that potential modulatory effects of NA on microglial function are mostly mediated by β2 receptor activation [90]. In line with findings in astrocytes, NA inhibited the IFNγ dependent upregulation of MHC-II antigens in microglial cells [92]. In addition, it has been shown that NA reduced NOS2 expression and the production of nitric oxide in inflammatory activated rat microglial cells, these actions being dependent on the activation of β2 adrenergic receptors and on the induction of IκBα production [93]. Inhibitory actions of NA were also detected using both mouse primary cultures of microglial cells and the murine N9 microglial cell line stimulated with the bacterial endotoxin, lipopolysaccharide (LPS) [94,95]. However, in the N9 microglia cell line, the inhibitory effect of NA on NOS2 activity was also dependent on the α adrenergic receptor activation, being mimicked by the α adrenergic agonist phenylephrine [94]. Interestingly, the exposure of rat microglial cells or mixed glial cultures to LPS alone or in combination with IFNγ downregulated the expression of β2 adrenergic receptors [96,97]. Similarly, the inhibition of LPS dependent NOS2 activity in microglia seemed to be in part mediated by α1 receptor activation [96]. In addition, NA effectively reduced the release of IL-1β [93,96,98], IL-6 [90,96] and tumor necrosis factor α TNFα [90,95,98] in rodent microglial cells activated with LPS, suggesting important anti-inflammatory effects. In contrast, direct stimulatory effects of NA on IL-6 release by both astrocytes and microglial cells were also described in the literature. Such effects are mostly mediated by activation of β2 adrenergic receptors [99]. Likewise, NE stimulated the release of IL-6, and other trophic factors, including GDNF and FGF2 by glial cultures in vitro [100]. Notably, IL-6 is a pleiotropic cytokine exerting relevant neurotrophic actions when released in low amount under basal conditions and with a crucial role in sustaining the inflammatory process and subsequent neuronal damage in several neurological disorders [101]. Conditioned medium (CM) derived from glial cells exposed to NA displayed trophic effects on cortical neurons in vitro, increasing the number of primary neurites extending from the cells, the number of neuritic branches and their length, leading to an enhanced Sholl profile in comparison to control CM. Interestingly, the direct application of NA had no effect on neuronal morphology. Moreover, the trophic effects of glial CM on neurons were reverted by inhibition of IL-6 receptors and neurotrophin signaling pathways [100], further suggesting that IL-6 may have a neuroprotective role under specific conditions. Of note, NA was found to increase IL-1β promoter activity, expression and synthesis in the mouse BV2 microglial cell line via activation of β2 adrenergic receptors [102], suggesting potential pro-inflammatory actions. Similarly, more recent studies also performed in primary microglial cultures stimulated with LPS demonstrated that NA potentiates the expression of cyclooxygenase 2 (COX-2) and the secretion of PGE2 induced by LPS [103]. Given the complexity of actions elicited by PGE2 on microglia through different receptors, its induction by NA could enhance the neurotoxic activities of these cells [104]. On the other hand, it has been shown that PGE2 can downregulate NOS2 expression through the activation of EP2 receptor [105]. Despite these conflicting results, the inhibition of NOS2 activity as well as the reduction of the release of proinflammatory cytokines, like IL1β, by microglial cells induced by NA exerted neuroprotective effects in neuronal cultures in vitro [96,106,107]. In addition, the neuroprotective actions of NA were also mediated by increased microglial release of anti-inflammatory mediators, such us the IL-receptor antagonist [108]. However, stimulatory effects of NA and isoproterenol on IL-1β release were observed in human microglial cells derived from post-mortem brain explants, cultured with GM-CSF and activated with LPS and nigericin. Notably, in these experiments NA was used at 10 µM whereas dopamine, at 1 mM, resulted in inhibitory actions [109]. Considering the relevant differences in microglial responses observed among different species [110], these recent data further highlight the importance of assessing the effect of NA on human microglial cells. In this regard the availability of human microglial cell lines may represent valuable experimental tools [111,112].

NA also increases the production of another protein that functions as a transcription factor such as peroxisome proliferator-activated receptor gamma (PPARγ). The activation of PPARγ reduces the expression of proinflammatory mediators in different cell types including glial cells [113]. The regulation of PPARγ by NA was observed in astrocytes and neurons and, as shown for IκBα, it seems also to depend on the activation of β adrenergic receptors [114].

### 5.2. Regulation of Microglial Activation by NA

Besides their production of different mediators, microglial activity can also be evaluated by assessing their morphology, motility or phagocytic activity. In this regard, it has been observed that NA causes a retraction of microglial processes in cell cultures and brain slices through the activation of β2 adrenergic receptors [115], suggesting that the activation of these receptors is an activation signal for microglia. In support of this, recent studies found that NA decreases the arborization of microglial cells, thus restricting the surveillance process, as well as the contact areas with neuronal dendrites [116,117]. Moreover, the addition of the β adrenergic agonist isoproterenol or the elevation of intracellular cAMP suppressed phagocytosis in primary microglial cultures [118]. However, several studies have shown that NA can significantly increase the microglial uptake of the Aβ 1-42 peptide, displaying a protective effect against AD lesions that is also mediated by activation of β2 adrenergic receptors [68,69,119]. These data confirm the complexity of the microglial activation process and the outcome of different signals probably depends on different conditions.

### 5.3. Neurotrophic Effects

NA has also been found to act directly on cultured neurons and, through the induction of nerve growth factor (NGF) and BDNF, reduce the lethal consequences resulting from the exposure of these cells to Aβ1-42 and Aβ25-35 peptides [120]. Remarkably, later in vivo analyses demonstrate how BDNF depletion in APP transgenic mice reduced the number of noradrenergic neurons in the LC while increasing the number of Aβ plaques [121]. The combination of these two studies suggests the existence of a reciprocal regulatory connection or a common regulatory system for NA and certain neurotrophic factors.

### 5.4. Regulation of Chemokines

In addition to the regulation of morphological changes and secretion of proinflammatory cytokines or neurotrophic factors, NA also seems to play a relevant role in the control of the production of certain chemokines by glial cells. We observed that CCL2, also known as MCP-1 or monocyte chemoattractant protein 1, is induced in cultured primary astrocytes by NA [122] and by noradrenaline transporter inhibitors or α2 adrenergic receptor antagonists [123]. As indicated by its common name, CCL2 is best known for its chemoattractant activity, that makes it a marker of inflammatory processes [124]. Therefore, CCL2 appears to be an interesting target for therapies aimed at reducing the progression of inflammation or migration of cells in certain situations such as cancer [125] or cardiovascular diseases [126]. However, CCL2 can also protect neurons against different types of injuries [127,128,129], excitotoxicity being one of the most studied ones [130]. In fact, we observed that NA induction of CCL2 by cultured primary astrocytes prevents the death of neurons caused by direct exposure to NMDA or glutamate. This CCL2 also protected neurons from glutamate neurotoxicity occurring when neurons were exposed to oxygen and glucose deprivation. Notably, this is an experimental model used to reproduce the conditions that take place in ischemic conditions in which excitotoxicity is the main cause of neuronal loss. These observations were further confirmed in vivo by elevating brain NA levels in mice using a NA precursor and a NA reuptake inhibitor [131].

However, while protecting neurons against excitotoxicity or other types of injuries, CCL2 may indirectly harm them through the stimulation of microglial cells and their subsequent production of neurotoxic agents. Interestingly, the analyses performed on microglial cultures indicate that, in the absence of other cells, CCL2 does not activate microglia because neither morphological changes nor the production of proinflammatory cytokines could be detected when the cultures were treated with different concentrations of CCL2 [132].

Based on these observations and on the existing publications demonstrating the contribution of CCL2 to certain neurodegenerative conditions, it was evaluated whether the activation state of glial cells could alter their responses to NA. In this regard, it has been shown that, in the presence of an inflammatory stimulus such as LPS, NA inhibits the production of CCL2 instead of inducing it as observed in control conditions [133]. The opposite effects of NA may also result from CCL2 regulation of the synthesis of β2 adrenergic receptors in astrocytes [134]. According to this, NA could contribute to maintain adequate levels of CCL2 in basal conditions but prevent its excessive accumulation when other stimuli have already elevated CCL2 concentration. This dual effect of NA is also in agreement with the neuroprotective actions of NA despite the induction of an inflammation-signaling chemokine. However, this may only occur as long as CCL2 is not already being released as a consequence of other factors.

CCL2, like many other mediators or signaling agents, can be detrimental when produced in excess, but the maintenance of certain constitutive levels may be required for homeostasis. As indicated above, controlling CCL2 has been demonstrated to protect neurons against different kinds of injuries. In fact, in 5xFAD mice, genetic removal of CCL2 reduced memory impairments, Aβ accumulation and neurodegeneration. In contrast, when older mice where analyzed, we could detect that CCL2 removal resulted in neuroinflammation and neuronal damage [135]. Therefore, more than a mere activating switch, NA seems to exert a complex control of CCL2 which could have deleterious consequences when it is lost as a result of LC degeneration.

As observed for CCL2, another chemokine known to play a relevant role in the control of glial activation such as CX3CL1 (also known as fractalkine) was found to undergo variable regulation by NA in astrocytes [133]. However, the neuroprotective actions of CX3CL1 have been studied in more detail than those of CCL2. Based on these actions, CX3CL1 has been proposed as a messenger used by neurons to communicate with microglia and restrain their activation [136]. In fact, we observed that neuronal secretion of CX3CL1 in response to NA reduced the production of nitrites in microglia [137]. Furthermore, the administration of the NA reuptake inhibitor reboxetine induced the expression of the CX3CL1 receptor CX3CR1 in the brain cortex of WT and 5xFAD mice [138]. CX3CR1 was also found to be elevated in human brain cortices from AD patients and in 5xFAD mice. This could be the result of an attempt to potentiate CX3CL1 anti-neuroinflammatory actions in pathogenic conditions. According to these observations, the lack of NA induction of CX3CR1 could contribute an increased degree of microglial activation. However, when isolated in culture, rat and human microglia displayed opposite effects, reducing the expression of CX3CR1 when treated with NA. This suggests that microglia responses to NA may also depend on the presence of the CX3CL1 produced by neurons or other factors of non-microglial origin. NA main mechanisms of action are summarized in Figure 2.

## 6. Targeting the Noradrenergic System in AD: New Potential Therapeutic Approaches

Since NA has an important role in regulating neuroinflammation and providing neuroprotection, NA-regulating drugs could contribute to the treatment of AD pathogenesis. Data from animal studies [13], as well as recent clinical trials, show a protective role of NA therapies and further confirm the anti-inflammatory and neuroprotective effects of NA previously demonstrated by in vitro studies (Table 3).

### 6.1. Noradrenaline Reuptake Inhibitor (NET)

Using rats in which the inflammation was elicited by the intraperitoneal injection of LPS, the administration of the NA reuptake inhibitors desipramine and atomoxetine reduced the expression of the pro-inflammatory cytokines IL-1β and TNFα as well as NOS2 and the microglial activation markers CD11b and CD40 in the brain cortex [139]. Moreover, at a later time desipramine and atomoxetine also reduced the expression of the chemokines interferon-inducible protein-10 (IP-10, CXCL-10) and regulated upon activation normal T-cell expressed and secreted (RANTES, CCL-5) as well as the cell adhesion molecules vascular cell adhesion molecule-1 (VCAM-1) and intercellular adhesion molecule-1 (ICAM-1) in cortex and also in hippocampus [140]. In our previous studies using the NET inhibitor reboxetine, we demonstrated that reboxetine treatment in 5xFAD mice model reduced neuroinflammation, amyloid burden and neurodegeneration [135]. In humans, treatment with atomoxetine in MCI patients reduced the levels of tau and phosphorylated tau and normalized biomarkers of synaptic function, brain metabolism, and glial immunity in CSF, and increased brain activity and metabolism in key brain circuits [141]. In addition, methylphenidate, a noradrenaline-dopamine reuptake inhibitor (NDRI), improves apathy and attention in AD patients [142].

### 6.2. Monoamine Oxidase Inhibitors (MAOI)

Monoamine oxidase inhibitors (MAOI) are the oldest type of antidepressants and an alternative strategy to elevate brain NA levels. For this reason, their administration can be expected to have some effect on the progression of AD. In addition, MAO activation has been linked to AD and activated forms of these enzymes are considered as biomarkers for this disease [143]. The accumulated experience in the prescription of certain MAOIs for the treatment of Parkinson´s disease, facilitated the assessment of their therapeutic potential in AD. However, while some studies have demonstrated positive effects with selegiline [144] or rasagiline [145], its clinical relevance in humans has not been confirmed yet [146].

### 6.3. α2 Adrenergic Antagonists

Presynaptic α2 adrenergic receptors mediate a negative feedback loop through which NA regulates its own secretion. Therefore, the administration of α2 adrenergic antagonists such as dexefaroxan also increases the release of NA in the CNS. This treatment promotes in rats the long-term survival of newborn neurons by reducing their apoptosis and increasing their complexity as well [147]. When a different α2 adrenergic antagonist such as fluparoxan was administered to APP/PS1 transgenic mice, it reduced memory deficits although no significant differences in Aβ accumulation or astrocytic activation were detected [148]. Treatment with indazoxan, a selective antagonist of α2AR, in APP/PS1 transgenic mice reduced Aβ40 and Aβ42 brain levels and amyloid deposition [149]. The induction of IL-10 has been proposed to contribute to the neuroprotective actions of these α2 adrenergic antagonists [150].

### 6.4. L-DOPS

Another strategy used to increase NA levels in the CNS is the peripheral administration of the NA precursor L-threo-3,4-dihydrophenylserine (L-DOPS) [151]. The administration of this compound to AAP-transgenic mice prevented the reduction of microglial migration towards Aβ plaques and their phagocytosis of these proteins [69]. Furthermore, L-DOPS reduced memory deficits and astrocyte activation in 5xFAD mice; it also induced the expression of various neurotrophic factors and Aβ degrading enzymes such as neprilysin and insulin degrading enzyme [152].

### 6.5. Tyrosine Hydrolase (TH) Potentiation

From a therapeutic point of view, it may seem more interesting to reduce LC neuronal damage or, at least, increase the activity of those neurons instead of just elevating NA levels or activity. For this purpose, vindeburnol, a semi-synthetic derivative of the plant alkaloid vincamine, is an interesting candidate molecule. Its intraperitoneal administration to rats increases the expression and activity of tyrosine hydroxylase, the rate limiting enzyme in NA synthesis [153]. Interestingly, vindeburnol re-activates the expression of tyrosine hydroxylase that is lost during development in certain LC neurons [154]. In 5xFAD mice, vindeburnol treatment reduced the accumulation of Aβ plaques, induced the expression of BDNF [155] and prevented some behavioral alterations characteristic of these mice [156].

### 6.6. LC Stimulation

An alternative way to potentiate the activity of LC neurons is through vagus nerve stimulation. This technique uses implantable devices which are connected to the vagus nerve and generate electrical impulses that function as signals transmitted through the nerve to certain brain areas, including the LC. Vagus nerve stimulation has demonstrated its efficacy in epilepsy, depression and some types of pain. For this reason, it was tested in AD patients although a rationale for its use based on a specific mechanism is not clearly established. The results for the first pilot study demonstrate that vagus nerve stimulation was well tolerated by the patients, its side effects were mild and transient and, more interestingly, it caused a significant improvement in two neuropsychological tests [157] that persisted one year later [158]. In rats, vagus nerve stimulation increases NA brain levels and the expression of BDNF [159].

While not tested in humans yet, potentiation of LC activity has also been performed in rats through chemogenetic activation via designer receptors exclusively activated by designer drugs (DREADD). The use of TgF344-AD rats is proposed to constitute a better strategy than other mouse-based models to reproduce the LC alterations found in AD due, among other factors, to the greater similarity between rat and human tau. Thus, learning impairments observed in transgenic rats were reduced when the LC was stimulated through DREADDs [160].

**Table 3 ijms-23-06143-t003:** Targeting noradrenergic system in preclinical models.

Authors and Year of Publication [Ref.]	Animal Model	Main Results of the Study
** *Noradrenaline reuptake inhibitor (NET)* **		
Gutiérrez et al., 2019 [135]	5xFAD Tg mice	Decreased inflammation, amyloid burden and neurodegeneration
O’Sullivan et al., 2010 [140]	Rats injected with LPS	Decreased inflammation
O’Sullivan et al., 2009 [139]	Rats injected with LPS	Decreased inflammation, T-cell activation and cells adhesion molecules
** *Monoamine oxidase inhibitors (MAOI)* **		
Tsunekawa et al., 2008 [144]	Mice injected with Aβ	Improved the cognitive impairments
** *α2 adrenergic antagonists* **		
Cheng et al., 2014 [149]	APP/PS1 Tg mice	Decreased amyloid deposition
Scullion el al. 2011 [148]	APP/PS1 Tg mice	Decreased deficits in spatial working memory
Rizk el al. 2006 [147]	Rats	Increased survival of newborn neurons
** *L-DOPS* **		
Kalinin et al., 2012 [152]	5xFAD Tg mice	Increased the expression of various neurotrophic factors and Aβ degrading enzymes
Heneka et al., 2010 [69]	APP V717F Tg mice injected with DSP-4	Restored microglia cells functions
** *Tyrosine hydrolase (TH) potentiation* **		
Braun et al., 2019 [156]	5xFAD Tg mice	Prevented behavioral alterations
Braun et al., 2014 [155]	5xFAD Tg mice	Decreased amyloid deposition and increased BDNF
** *LC stimulation* **		
Rorabaugh et al., 2017 [160]	TgF344-AD rats	Reduced learning impairments
Follesa et al., 2007 [159]	Rats	Increased NA, BDNF and bFGF

## 7. Detrimental Effects of NA in Alzheimer’s Disease

Despite all the beneficial and neuroprotective actions of NA described above, several studies demonstrate the existence of negative effects mediated by NA in AD. Obviously, excessively elevated levels of NA could be detrimental.

### 7.1. Higher Levels of NA Can Worsen Cognitive Functions

It is known that stress exposure can increase NA release and excessive brain levels of NA can contribute to cognitive deficits including memory impairments [161]. The compensatory mechanism that takes places in AD could lead to higher NA levels increasing noradrenergic signaling [38]. In fact, higher levels of NA in CSF correlated with cognitive deficits [37]. In addition, an excess of NA could lead to increased production of toxic metabolites, as mentioned above, contributing to LC damage and AD progression [162]. Thus, the contribution of noradrenergic system dysfunction to AD pathology and progression is not fully elucidated, and it is also possible that NA-elevating drugs administered in the early stages of AD can result in detrimental effects, worsening the cognitive symptoms.

In addition, due to the changes in distribution and function of the different adrenergic receptors in AD patients, increases in NA brain levels could have some deleterious effects, and its inhibition could also be considered a therapeutic target.

### 7.2. Role of Adrenergic Receptors in Amyloid Deposition

The compensatory upregulation of βAR that takes place in the early stages of AD could increase the amyloid deposition due to the role that these receptors play in amylogenesis. One study used the β adrenergic agonist isoproterenol to stimulate HEK293 and primary hippocampal cultures and observed increased Aβ production and γ-secretase activity in both cell types. In this study, NA was also administered to rats through intracerebroventricular injections, causing an enhancement of γ-secretase activity and Aβ production in the hippocampus. In support of this, chronic administration of isoproterenol or clenbuterol (a specific β2 adrenergic receptor agonist) to APP/PS1 mice increased the density of amyloid plaques, while the β2 adrenergic receptor antagonist ICI 118,551 had the opposite effect [40].

As a complement to these results, the opposite strategy, based on the use of a β adrenergic receptor antagonist such as propranolol, was found to reduce the accumulation of Aβ in neuronal cultures derived from the APP overexpressing Tg2576 mice [163]. Furthermore, in vivo analyses performed on a mouse line that displays a phenotype of accelerated aging, known as senescence accelerated mouse prone 8 (SAMP8), demonstrate that propranolol administration for three weeks attenuates cognitive memory impairments and reduces Aβ and hyperphosphorylated tau accumulation [164]. This research group found similar results in Tg2576 mice in which propranolol in addition to the aforementioned memory, Aβ and tau effects, also enhanced the expression of insulin degrading enzyme, suggesting this way a potential mechanism for propranolol Aβ-reducing activity [165]. In addition, treatment of Tg2576 mice with nebovilol, a β1 adrenergic receptor antagonist, reduces Aβ production [166].

Due to the widespread use of β blocking drugs to control hypertension, a large amount of data is available, and this has permitted several population-based studies recruiting patients treated for relatively long periods of time with this category of drugs. In this way, combining the results of dementia evaluation tests and the pharmacological history of patients, β blocking drugs were observed to slow the progression of functional decline as assessed by a Clinical Dementia Rating [167]. Additionally, a study focused on β blockers treatment for three years prior to the development of AD, found that β blocking treatment is associated with reduced odds of developing dementia [168].

On the other hand, α adrenergic receptors could also increase Aβ production and mediate Aβ toxic effects. Treating APP/PS1 mice with clonidine, a α2 adrenergic agonist, increases Aβ deposition, and the use of the α2 adrenergic antagonist idazoxan decreases Aβ load improving the cognitive deficits [149]. In addition, Zhang et al. demonstrated that Aβ oligomers can bind to an allosteric site on α2AAR [169]. The binding of amyloid oligomers redirects the pathway to glycogen synthase kinase 3β (GSK3β) activation, the main enzyme that phosphorylates tau, increasing the levels of tau hyperphosphorylation. Furthermore, treatment of APP23 mice with prazosin, an antagonist of α1AR, rescued the cognitive deficits while increasing anti-inflammatory cytokines [170], and the treatment with prazosin reduced the agitation and aggressive symptoms of AD patients [171].

Thus, the malfunction of adrenergic receptors due to LC loss could contribute to Aβ deposition and toxicity contributing to AD progression, and its inhibition could be also considering as a therapeutical approach.

## 8. Conclusions

The clear relationship between the loss of noradrenergic neurons and the development of AD indicates that NA alterations are associated with this pathology. In addition, the use of different animal models based on the reduction of NA production through different means, confirmed the causal effect of this alteration and the memory deficits, neuroinflammation and neurodegeneration characteristic of AD.

In vivo MRI analysis of LC could be used as an early biomarker to diagnose and monitor AD. In addition, LC dysfunction by MRI analysis should be considered a promising target for the identification of at-risk individuals in the prodromal stages of AD, and for the development of novel preventive interventions that increase NA signaling such as cognitive-attention training, physical exercise, diet, and therapeutic interventions.

In addition to the harmful effects of LC deterioration, the studies focused on the analysis of NA actions have demonstrated that this neurotransmitter has a clear anti-inflammatory and neuroprotective action in the brain. Therefore, the maintenance of NA normal endogenous levels seems to be a relevant therapeutic goal in all situations in which neuroinflammation threatens brain homeostasis. The existence of different NA-elevating drugs, currently approved and used for depression and other neurological disorders, eliminates the need to develop new substances as well as the long and costly process required to permit their administration to patients. This way, the repurposing of the aforementioned drugs could be an interesting alternative to current pharmacological treatments for AD.

Figure 3 includes a summary of pharmacological tools that can be used to regulate NA levels.

## 9. Future Directions

As we previously described, elevating NA levels represents a promising therapeutic opportunity in AD. However, further research is still needed to unravel the full contribution of noradrenergic system dysfunction in AD progression and confirm the NA-related drug therapeutic effects in AD patients.

We need to better characterize the changes in the LC system that occur during AD progression. Both noradrenergic hyperactivity and a decrease of its activation could contribute to AD pathogenesis. Therefore, it is mandatory to determine the detrimental and/or beneficial effects of LC activation at the different stages of AD. This could help us to establish the best timeframe for using NA-related drugs in combination with other treatments and to predict their effectivity.

Finally, future studies using humanized research tools are imperative to further confirm if NA actions and its therapeutic effects observed in animal models could also be translated to humans.

## Figures and Tables

**Figure 1 ijms-23-06143-f001:**
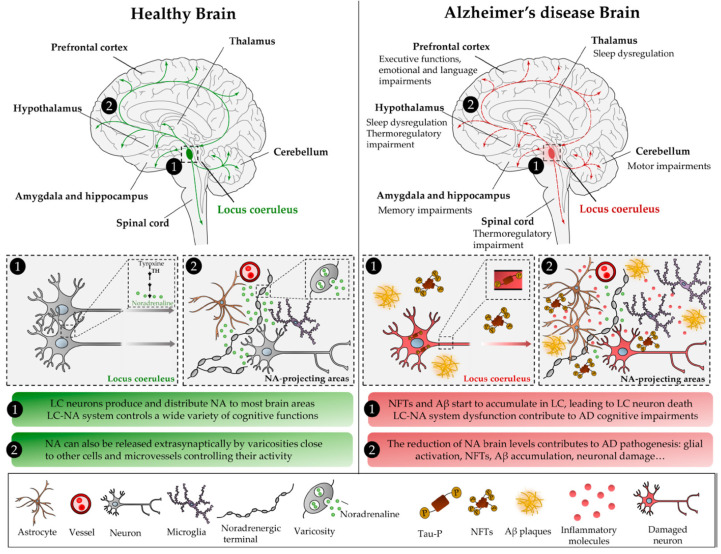
Main alterations in the CNS resulting from locus coeruleus degradation in Alzheimer’s disease.

**Figure 2 ijms-23-06143-f002:**
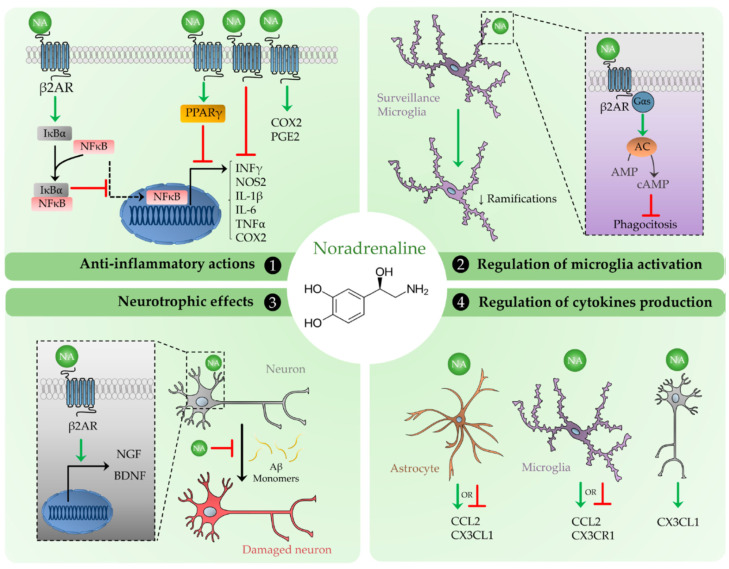
Neuroprotective actions of noradrenaline.

**Figure 3 ijms-23-06143-f003:**
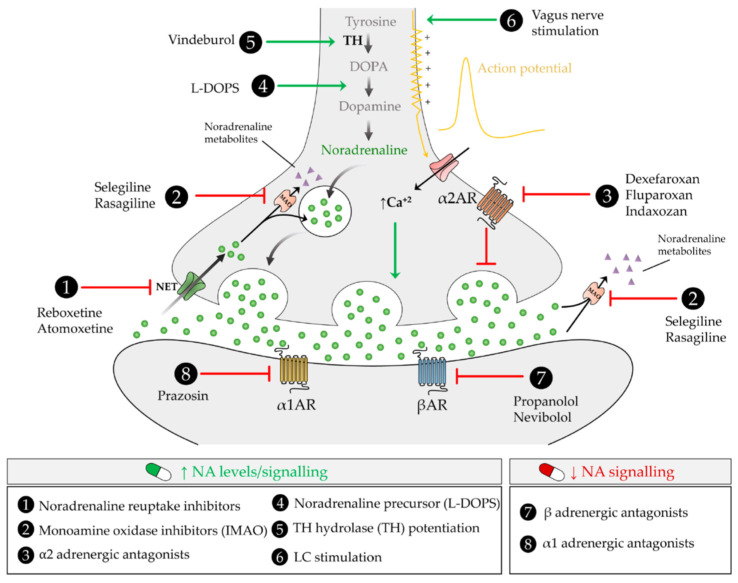
Graphic representation of some of the best known exogenous modulators of brain NA levels.

## Data Availability

Not applicable.

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
