# Peer review of "Noradrenaline in Alzheimer’s Disease: A New Potential Therapeutic Target"

_ijms, 2022, doi:10.3390/ijms23116143_

Round 1

Reviewer 1 Report

This review summarizes the relevant literature describing the alterations of the noradrenergic system in Alzheimer´s disease and experimental studies linking noradrenaline function with alterations affecting the behavior and viability of different nervous cells.

Combination of the multiple studies indicate that the maintenance of adequate noradrenaline levels in the central nervous system is critical endogenous defense system that help preventing Alzheimer´s disease.

The authors cold try to improve their text and to add a scheme summarizing the multiple mechanisms referred in the manuscript and how they are linked each other. A comment on pre-symptomatic stages such as Mild Cognitive Impairment and noradrenaline should also be included.

A graphical abstract at the beginning might help reader to navigate through this text

Reviewer 2 Report

The manuscript entitled “Noradrenaline in Alzheimer´s disease: a new potential therapeutic target” summarizes some of the most relevant data describing the alterations of the noradrenergic system typically occurring in Alzheimer´s disease, as well as experimental studies in which noradrenaline concentration was modified to analyze further how these alterations affect the behavior and viability of different nerve cells. The combination of the various studies here presented suggests that the maintenance of adequate noradrenaline levels in the central nervous system constitutes a key factor of the endogenous defense systems that help prevent or delay the development of Alzheimer´s disease.  The review is informative and well written. The only recommendation is to thoroughly check the spelling and grammar to maintain uniformity (font and size) throughout the manuscript, including the schemes and figures.
